# ROYAL SOCIETY
# OPEN SCIENCE

optics/biogeochemistry/climatology

lidar, satellite, global, continuous coverage, vegetation mapping

**Author for correspondence:**
Steven Hancock
e-mail: steven.hancock@ed.ac.uk

# Requirements for a global lidar system: spaceborne lidar with wall-to-wall coverage

Steven Hancock[1], Ciara McGrath[2], Christopher Lowe[2], Ian Davenport[1] and Iain Woodhouse[1]

[1]School of Geosciences, University of Edinburgh, Crew Building, Edinburgh EH9 3FF, UK
[2]Applied Space Technology Laboratory (ApSTL), Department of Electronic and Electrical Engineering, University of Strathclyde, 204 George St, Glasgow G1 1XW, UK

  SH, 0000-0001-5659-6964; CM, 0000-0002-7540-7476

Lidar is the optimum technology for measuring bare-Earth elevation beneath, and the structure of, vegetation. Consequently, airborne laser scanning (ALS) is widely employed for use in a range of applications. However, ALS is not available globally nor frequently updated due to its high cost per unit area. Spaceborne lidar can map globally but energy requirements limit existing spaceborne lidars to sparse sampling missions, unsuitable for many common ALS applications. This paper derives the equations to calculate the coverage a lidar satellite could achieve for a given set of characteristics (released open-source), then uses a cloud map to determine the number of satellites needed to achieve continuous, global coverage within a certain time-frame. Using the characteristics of existing in-orbit technology, a single lidar satellite could have a continuous swath width of 300 m when producing a 30 m resolution map. Consequently, 12 satellites would be needed to produce a continuous map every 5 years, increasing to 418 satellites for 5 m resolution. Building 12 of the currently in-orbit lidar systems is likely to be prohibitively expensive and so the potential of technological developments to lower the cost of a global lidar system (GLS) are discussed. Once these technologies achieve a sufficient readiness level, a GLS could be cost-effectively realized.

## 1. Introduction

Light detection and ranging (lidar) has been shown to be the optimum technology for accurately mapping ground elevation through vegetation and in steep terrain [1,2]. Because of this, airborne lidar has been widely employed to make bare-Earth topographic maps for use in flood modelling, to understand geomorphology and to investigate archaeological sites [3,4]. In addition, lidar can measure the vegetation structure above the

ground, allowing it to map vegetation properties including carbon content [5], fine scale habitat structure [6], ecosystem services [7] and microclimates [8].

Past studies have shown that lidar does not suffer the signal saturation over dense vegetation suffered by passive optical [9] or radar [10,11] observations, and so lidars can make accurate measurements of the densest forests on the Earth [12]. Lidar's short wavelength light source (532–1550 nm) and the very narrow beam (compared with radar), allows a direct retrieval of high-resolution surface and vegetation height, down to the centimetre level.

While radars can have high precision over hard targets [13], they cannot achieve the same resolution and accuracy as lidar in the presence of vegetation because their longer wavelengths do not scatter from the individual elements, nor pass through the small gaps within the canopies without interacting, leading to less direct measurements of the targets [11,14]. Furthermore, while global radar elevation products are available, they all suffer from the vegetation bias resulting from the scattering from the canopy elements.

Photogrammetric techniques, such as structure from motion [15], can measure high-resolution three-dimensional structure under some circumstances, but this technique relies upon unobstructed views of recognizable objects from multiple directions. In areas without large gaps between vegetation (any partially closed canopy or dense understorey), the method struggles to identify the ground beneath vegetation [16]. Lidar does not suffer from these problems, and it is lidar's ability to penetrate dense vegetation cover that makes it the optimum technology for measurements in vegetated environments.

Currently, the majority of lidar data are collected by airborne laser scanning (ALS), which achieves the high detail and accuracy expected from lidar, but has a high cost per unit area. As an illustration, Wales has recently contracted for complete ALS coverage at a cost of £2 million [17] (approx. $2.5 million) for a density of 2 points $m^{-2}$. This is £96 $km^{-2}$. Scaling that up to the global land area would require £42 billion for a single acquisition (ignoring possible savings from economies of scale but also ignoring additional costs for surveying less accessible regions). It should also be noted that denser forests require higher point density for accurate measurement [18]—around 4–10 points $m^{-2}$. An argument has been made for a global ALS survey and $250 million cited [19], but that proposal suggested a sampled subset of global forest cover, not the entire land surface. While such an approach may be sufficient to improve estimates of global forest carbon stock at the regional scale, it is not necessarily suitable for the many other applications that lidar is suited for, such as flood risk mapping or in heterogeneous areas such as cities or commercial forestry. For this reason, the current paper's focus is on achieving global, continuous coverage.

Owing to their high cost, ALS datasets are typically collected infrequently, over limited areas and generally only within wealthy countries. This prevents the routine application of ALS data globally and limits its use in monitoring changes over time. Satellite lidar can provide global coverage at a fraction of the cost (for example, $94 million for GEDI for all land between 51.6° N and S), but current spaceborne lidars have very sparse sampling (GEDI will directly measure less than 4% of the Earth), preventing their use in applications that require continuous coverage or low sampling errors for change detection. Here we calculate how spaceborne lidar technology could be scaled to create a system with continuous coverage, enabling applications that previously relied on ALS data.

## 1.1. Existing spaceborne lidars

To date there have been six Earth observation lidar missions, listed in table 1, in addition to the technology demonstrator LITE mission that was carried on the Space Shuttle [20]. These missions were designed for a range of applications. ICESat, carrying the GLAS instrument and ICESat-2, carrying the ATLAS instrument, were designed to measure ice-cap volume [21,22]. CALIPSO, carrying the CALIOP instrument and CATS, mounted on the International Space Station (ISS), were designed to measure clouds and aerosols [23,24]. Aeolus, carrying the ALADIN instrument, was designed to measure clear air wind speed [25]. GEDI, mounted on the ISS, was designed to measure forests [2]. These different applications required slightly different designs, but all instruments share some common aspects and limitations.

All of these spaceborne lidars collect unique data. For example, Aeolus is the only instrument capable of remotely detecting wind speed in clear air [25] and GEDI will arguably produce the most accurate, non-saturating biomass map of the tropical rainforests [2]. However, the fundamental limitation of spaceborne lidar is that it is an active remote sensing technique and must provide its own energy for illumination. This energy requirement limits the area that can be covered compared with other remote sensing technologies (since every point measured must be illuminated first). Passive optical instruments collect energy emitted from the sun or the target, while radars use longer, lower energy per photon, wavelengths. Lidars sample very small areas compared with both passive and radar sensors and must

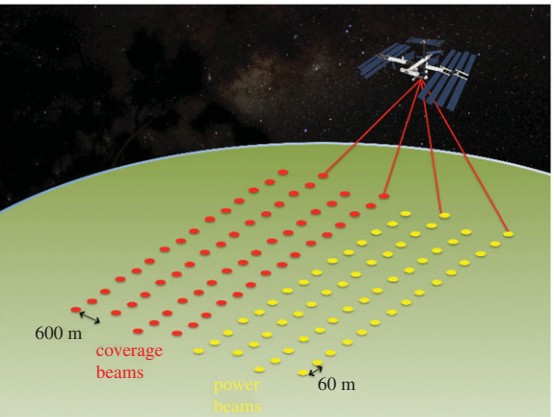

**Figure 1.** Illustration of GEDI sampling.

**Table 1.** Characteristics of spaceborne lidars flown to date. Note that ICESat-2's 1.2 mJ output laser energy is subsequently split into six beams by a diffractive optical element. GEDI has three 10 mJ lasers firing simultaneously, one of which is split into two 5 mJ beams. CATS had two lasers, one firing 2 mJ pulses at 4 kHz and the other firing 1 mJ pulses at 5 kHz. All other instruments emit a single laser beam. All lasers operate at 1064 nm with some frequency doubled or tripled to 532 nm or 355 nm.

| mission | pulse rate | energy per shot | output power | telescope area | wavelength | operated |
|---|---|---|---|---|---|---|
| ICESat | 40 Hz | 100 mJ | 4 W | 0.50 m$^2$ | 1064/532 nm | 2003–2009 |
| CALIPSO | 20 Hz | 110 mJ | 2.2 W | 0.79 m$^2$ | 1064/532 nm | 2006– |
| CATS | 10 kHz and 5 kHz | 1 mJ and 2 mJ | 15 W | 0.28 m$^2$ | 1064/532/355 nm | 2015–2017 |
| Aeolus | 51 Hz | 110 mJ | 5.6 W | 1.8 m$^2$ | 355 nm | 2018– |
| ICESat-2 | 10 kHz | 1.2 mJ | 12 W | 0.50 m$^2$ | 532 nm | 2018- |
| GEDI | 242 Hz | 3 × 10 mJ | 7.3W | 0.50 m$^2$ | 1064 nm | 2018- |

use multiple beams to sample across a swath. Existing systems have operated with between one (ICESat, CALIPSO and Aeolus) and eight (GEDI) narrow ground tracks from a single overpass.

To illustrate this difference in coverage, the passive optical Landsat-9 measures a continuous swath 185 km wide on a single overpass [26] to give global coverage every 16 days, while the radar Sentinel-1 measures a continuous swath of 290 km [27] to give global coverage every 6 days. In comparison, GEDI has the widest spaceborne lidar swath and the densest across-track coverage, with eight tracks of 22 m diameter beams, each separated by 600 m across-track, for a total width of 4.2 km, illustrated in figure 1. During its planned 2 year mission, GEDI's sparse sampling will result in only around 2–4% of the Earth's land surface being directly measured [2].

Owing to their sparse sampling, lidars require either the sampling error to be accepted, making use of statistical techniques to calculate it [28], or fusion with continuous datasets and/or models to make use of the sparse, but accurate, lidar data to calibrate a continuous, but less accurate dataset [29,30]. While these fusion methods can reduce the sampling error, the resulting products are not as accurate as a full-cover lidar product [31]. Meanwhile, the products from lidar-only statistical methods are too coarse to be applied to flood modelling, commercial forestry (where parcels of land are small) or urban mapping (where features change at high spatial frequency). These three applications are often the primary economic reasons for collecting ALS data, and so spaceborne lidar is not yet a substitute for ALS surveys.

There are a number of proposed future spaceborne lidars with similar characteristics to those in table 1, such as JAXA's MOLI [32] and CNES's LEAF [33], but one Earth orbiting lidar mission has been proposed with much greater coverage than the existing missions: NASA's LIST [34]. If realized, LIST would have a 5 km wide continuous swath made up of 1000 beams illuminating 5 m footprints, allowing it to provide continuous global coverage within about 3 years (the time for complete coverage was not reported in the study of Yu *et al.* [34] and was estimated from the method described in §2.4). Each footprint would contain 100 µJ of laser energy, a much lower energy per shot than the instruments in table 1 and so new algorithms or more efficient detectors would be needed to extract

information with sufficient accuracy. The total output laser power would be 1 kW (1000 times 100 μJ pulsing at 10 kHz [34]), more than 80 times the output power of the current most powerful spaceborne lidar. This would be a demanding engineering task, even if laser efficiencies were raised from the current in-orbit limit of 5–8% [35,36] to the proposed 15% [34] and would require 6.7 kW of power for the instrument (doubling to 13.4 kW to account for heat dissipation). Furthermore, few details have been made public since the description of the airborne simulator [34] was published in 2010. There are also proposals for wide coverage extra-terrestrial lidars [37], but the lower orbits, thin atmospheres and lack of vegetation cover on Mars, Mercury and the Moon make those easier to engineer than for the Earth.

## 1.2. Research questions

For a spaceborne lidar to be applied to uses which have previously relied on expensive ALS data, it must achieve continuous coverage. For a global lidar system (GLS) with continuous coverage, each pulse must have sufficient energy to make an accurate measurement (i.e. see the ground clearly above background noise, even when penetrating a dense canopy) and every point on the Earth at a given spatial resolution must be illuminated by a laser pulse within a specified time-frame.

To calculate how the energy requirements of existing spaceborne lidars must be scaled to make a GLS, the following research questions must be answered:

Section 2.1: Estimate minimum return energy per laser pulse.
Section 2.2: Estimate achievable swath width for a given output power.
Section 2.3: Find maximum current output power per platform.
Section 2.4: Calculate number of lasers needed for global coverage.
Section 2.5: Estimate rough total cost of system.

This paper will answer these questions in the following sections and some suggestions will be made for technological developments that may allow a GLS to be realized more cost-effectively in §3. It should be noted that a lidar simulation tool capable of answering these questions was reported by Paige [38,39], but no further publications seem to have been released since 2017, and no code, sufficient details to implement, or results are available.

# 2. Global lidar system requirements

The number of footprints, and so laser pulse rates and power, is controlled by the required resolution, which in turn is controlled by the laser footprint size. The resolution requirement depends upon the application but there are some limits. It has been shown that too large a footprint will prevent the ground being seen underneath a forest due to overlap between the ground and vegetation returns [40,41]. For example, ICESat cannot be used to measure forest height on slopes steeper than 10–12° [42] because of its large (65–90 m diameter) footprint. An upper footprint diameter limit of around 30 m has been suggested [2]. There is no lower limit to the resolution, until the diffraction limit (which, for example, is 65 cm for GEDI's 0.5 $m^2$ telescope at an altitude of 405 km using a 1064 nm wavelength), but as the laser footprint gets smaller, the number of pulses needed to achieve the sampling density required for higher resolution increases, and so the total power requirement is greater. The National Research Council's decadal survey has set a minimum resolution of 5 m for a bare-Earth digital elevation model (DEM) [43], but it should be noted that a global bare-Earth DEM of even 30 m resolution would revolutionize a number of fields. Some applications require very high spatial resolution, such as detecting small dwelling foundations for archaeology and mapping street furniture [4]. These tend to require multiple (10+) pulses per square metre, which would require resolutions of a few tens of centimetres and so be very challenging to achieve with a satellite lidar given the diffraction limit and the technology likely to be available over the next few years. In addition, at those spot sizes the system would suffer significant geolocation issues from laser jitter [44]. As many applications would benefit from 5 to 30 m resolution data, especially vegetation studies [2], flood modelling [45] and some geomorphological processes [46], this paper will focus on systems with the less energy intensive spatial resolutions of 5–30 m.

For the minimum temporal resolution, there has never been a global bare-Earth DEM, so any time-scale is an improvement over what is currently available. For those countries that have continuous ALS coverage, the data have not been collected more regularly than once every 10 years, so this could provide an upper limit to the time to coverage.

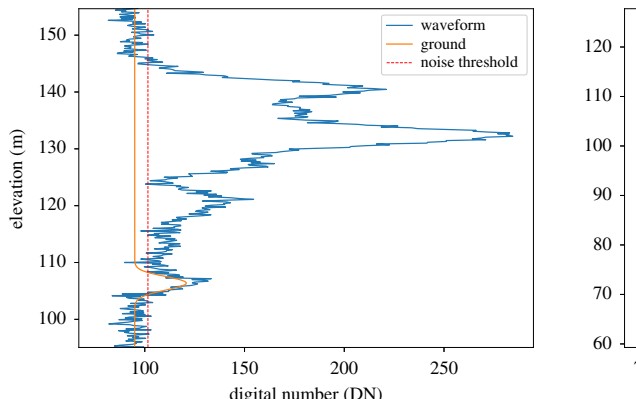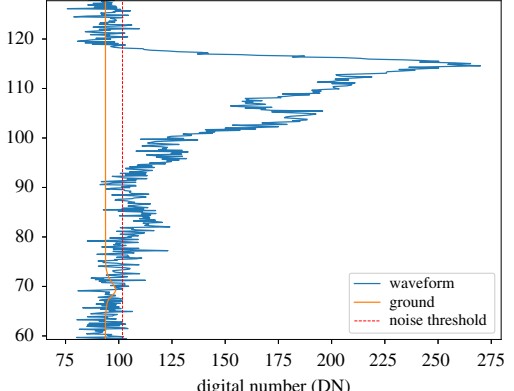

**Figure 2.** Illustration of minimum detectable energy for a lidar return with a 98% beam sensitivity over areas with 95% (left) and 99% canopy cover (right). The lidar waveform, including noise, is shown by the blue line. The ground part of that waveform is shown by the orange line. The dotted red line shows the threshold needed to reliably remove background noise for this signal-to-noise ratio (SNR). If the beam sensitivity is less than the canopy cover, the ground return will be beneath that threshold and so likely to be undetectable.

This paper will investigate the requirements for producing a global map with between 5 and 30 m resolution, completed once every 1–5 years. Five years has been chosen rather than 10 as satellite lidars have so far only been designed for 3–6 years of operation, although CALIPSO has far exceeded that. It should be noted that the equations presented can be used for any spatial-resolution product and the preliminary analysis did consider 1 m resolution, but the number of spacecraft required were already so large at 5 m resolution that the results are only presented for 5–30 m resolution.

## 2.1. Energy per pulse

For a remote sensing lidar capable of measuring bare-Earth elevation and the structure of objects above, there must be sufficient signal returned from the ground to detect it above background noise. This is illustrated in figure 2. The signal-to-noise ratio (SNR) can be defined in terms of the beam sensitivity, using eqn (6) in Hancock *et al.* [47]. This gives the canopy cover at which the SNR of the ground return is such that there is a 90% chance of correctly detecting it above a noise threshold. If the vegetation canopy cover is greater than the beam sensitivity, there is a high probability of an inaccurate ground estimate and so the denser the forest, the higher the SNR needs to be for an accurate measurement. If the ground return is not detected, the estimate of ground elevation and the height of targets will be inaccurate.

The SNR of a lidar signal is controlled by the amount of laser energy returned from the target and the amount of background noise (from background light and electronic noise). Using GEDI as an example, which is the only spaceborne lidar specifically designed to measure the ground elevation under dense vegetation, it was calculated that a 10 mJ laser pulse would be able to detect the ground through a canopy cover of 99.75% by night (when background noise is low) and 97% by day (when background noise is highest) while a 5 mJ pulse would be able to detect the ground through a canopy cover of 98% by night and 96% by day [2,47]. This is true for GEDI's case of a 1064 nm laser, 0.5 m$^2$ telescope [2], 45% detector efficiency [48] and altitude of 405 km, assuming 80% mean atmospheric transmittance, a bare-Earth reflectance of 0.4 and a vegetation reflectance of 0.57. Note that the beam sensitivities reported in Hancock *et al.* [47] applied a conservative link margin loss in case the pre-flight predicted performance was not achieved.

To generalize this for any altitude, telescope size and detector efficiency, the laser pulse energy, $E_{shot}$, can be related to the energy absorbed by the detector, $E_{det}$ by calculating how much energy is lost to absorption by the surface, scattering by the atmosphere and how much is collected by the detector through the lidar equation.

$$E_{shot} = \frac{E_{det}}{Q} \frac{2\pi h^2}{A} \frac{1}{\rho \tau^2},\qquad(2.1)$$

where $Q$ is the detector efficiency, $A$ is the telescope area, $h$ is the instrument altitude, $\rho$ is the surface reflectance and $\tau$ is the atmospheric transmittance. Solving equation (2.1) in terms of $E_{det}$ for GEDI's characteristics gives the return energies and associated beam sensitivities by day and night listed in table 2. This shows that GEDI's 10 mJ 'power' beam (beam sensitivity of 99.75% by night and 97% by

**Table 2.** Return energy absorbed by the detector, $E_{det}$, needed to reliably detect the ground for different canopy covers and background light conditions, provided in femto-Joules and number of photons (at 1064 nm).

| beam sensitivity | night/day | $E_{det}$ | return photons |
|---|---|---|---|
| 96% | day | 0.281 fJ | 1505 |
| 98% | night | 0.281 fJ | 1505 |
| 97% | day | 0.562 fJ | 3009 |
| 99.75% | night | 0.562 fJ | 3009 |

day) has a detected energy of 0.562 fJ and the 5 mJ 'coverage' beam (beam sensitivity of 98% by night and 96% by day), has a detected energy of 0.281 fJ.

Thus for a GLS with a mean beam sensitivity of 98% by night (and 96% by day), the satellite must be configured to provide a mean detected energy of 0.281 fJ per shot. Note that these minimum detected energies ($E_{det}$) are for the ground return to be clearly identifiable above background noise in a single footprint, which is essential for detecting the ground in isolated lidar footprints such as GEDI's. For a lidar system with continuous coverage, using information from adjacent footprints may allow the required energy per shot to be reduced [49,50].

## 2.2. Swath width

Now that a minimum laser pulse energy has been identified in table 2, this can be combined with the pulse repetition rate (PRR) to give the total output laser power needed. This can then be related to the satellite payload power to determine how wide a swath of footprints can be emitted from a single satellite.

The laser pulse rate needed to achieve continuous coverage can be calculated as the satellite ground speed divided by the required sampling distance (the ground resolution), $r$. For circular orbits, ground speed is constant for a given orbit altitude, $h$, and the PRR, $\Psi_0$, for a single ground track (one strip of pixels), is

$$\Psi_0 = \frac{R\sqrt{GM}}{r(R+h)^{3/2}}, \tag{2.2}$$

where $R$ is the radius and $M$ the mass of the Earth, and $G$ the gravitational constant. To obtain a swath width, $s$, the total pulse rate, $\Psi$, on the satellite must be multiplied by the ratio of the swath width to resolution, $r$

$$\Psi = \Psi_0 \frac{s}{r}. \tag{2.3}$$

The total laser energy requirement is then $\Psi$ multiplied by the energy per laser pulse, $E_{shot}$ in equation (2.1). Note that $\Psi$ in equation (2.3) is the rate of pulses leaving the satellite rather than the pulse rate for an individual laser. These separate pulses could be generated by splitting a single laser pulse into multiple output beams, as used on ICESat-2 [22], by having multiple lasers firing simultaneously, as used on GEDI [2], scanning the laser across-track and increasing the PRR as used on ALS, or some combination of the three. The total power requirement will be the same in all cases.

The total output power is limited by the power the satellite platform can provide to the instrument payload, $P_{pay}$ and the laser efficiency, $L_e$, which includes energy used for thermal management. Note that the product of $P_{pay}$ and $L_e$ gives the total laser output power, shown in table 1.

$$\Psi E_{shot} \leq P_{pay} L_e. \tag{2.4}$$

Combining equations (2.1)–(2.4), the swath width a lidar satellite can achieve is defined as a function of satellite payload power, $P_{pay}$, laser efficiency, $L_e$, detected energy needed, $E_{det}$, altitude, $h$, resolution ,$r$, telescope area, $A$, detector efficiency, $Q$, surface reflectance, $\rho$, and atmospheric transmissivity, $\tau$.

$$s = \frac{P_{pay}L_e}{E_{det}} \frac{A}{2\pi h^2} Q\rho\tau^2 \frac{r^2(R+h)^{3/2}}{R\sqrt{GM}} \tag{2.5}$$

Equation (2.5) shows that swath width increases with decreasing altitude, and increasing telescope area and laser and detector efficiency. While a lower altitude is beneficial in terms of swath width, the

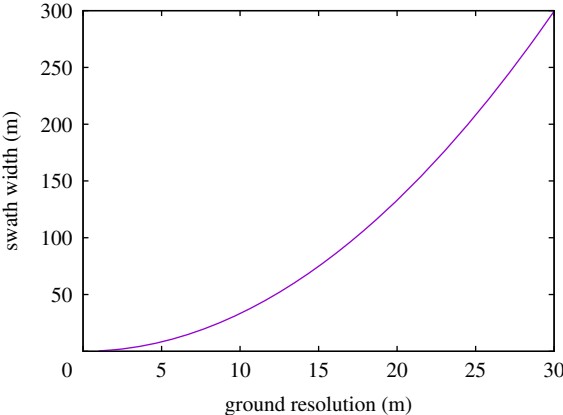

**Figure 3.** Swath width achievable against ground resolution for a 12 W lidar satellite in a 405 km orbit with a 0.5 m$^2$ telescope.

associated increase in atmospheric drag will require a greater volume of propellant to maintain the orbit altitude for the duration of the mission. As a compromise, and to offer direct comparison with an existing mission, GEDI, an altitude of 405 km is used for the rest of this paper as a minimum value.

## 2.3. Output power per platform

From table 1, the current highest output power free-flying lidar satellite is ICESat-2 at 12 W (CATS is mounted on the ISS). Note that higher output powers than this may be possible [34,37], but for this paper the highest demonstrated lidar output power will be used as an upper limit. Using the $P_{pay} \cdot L_e = 12$ W from ICESat-2 and $E_{det} = 0.281$ fJ, $A = 0.5$ m$^2$, $Q = 45\%$, $h = 405$ km, $\tau = 0.8$ from GEDI, the swath widths for different resolutions are shown by the curve in figure 3. $\tau = 0.8$ is chosen as an average value to give an average returned energy and so the average beam sensitivity of 98% by night. It should be noted that atmospheric transmission varies in space and time, but this paper explores what constellation would be required to maintain a mean beam sensitivity of 98% by night, echoing the method used for the planning for GEDI [2,47]. The beam sensitivity will vary as the atmospheric transmission varies and some additional data may require filtering out if the beam sensitivity falls below the local canopy cover, although a continuous raster of footprints should allow that to be offset by more advanced signal processing [49,50].

It can be seen that the achievable swath width rapidly decreases as resolution is made finer. For some configurations, the swath width becomes narrower than the resolution. In these cases, the satellite is not capable of providing sufficient power for even a single continuous transect. From this, the minimum possible resolution that can produce a continuous track, $r_{min}$, for a given platform power, can be derived by substituting $\Psi$ in equation (2.4) with $\Psi_0$ from equation (2.3).

$$r_{min} = \frac{E_{shot}}{P_{pay}L_e}\frac{R\sqrt{GM}}{(R+h)^{3/2}}.$$ (2.6)

A 12 W output lidar satellite, requiring 0.281 fJ of detected energy, at an altitude of 405 km, with a 0.5 m$^2$ telescope, would be limited to a finest resolution of 3 m. To achieve a finer spatial resolution would require a larger telescope, accepting less energy per laser pulse (as used on ICEsat-2 [22]), a higher output power or a lower orbit.

The next section will combine this finding with orbital simulations to calculate how many platforms would be required to achieve global coverage for different spatial and temporal resolutions.

## 2.4. Number of spacecraft required

### 2.4.1. Cloud data loss

Clouds block lidar measurements so there need to be sufficient overpasses of a satellite to give a good probability of at least one cloud-free observation. The probability of at least one cloud-free observation, $p_{obs}$, can be calculated as one minus the probability of no observations given a number of

(a)

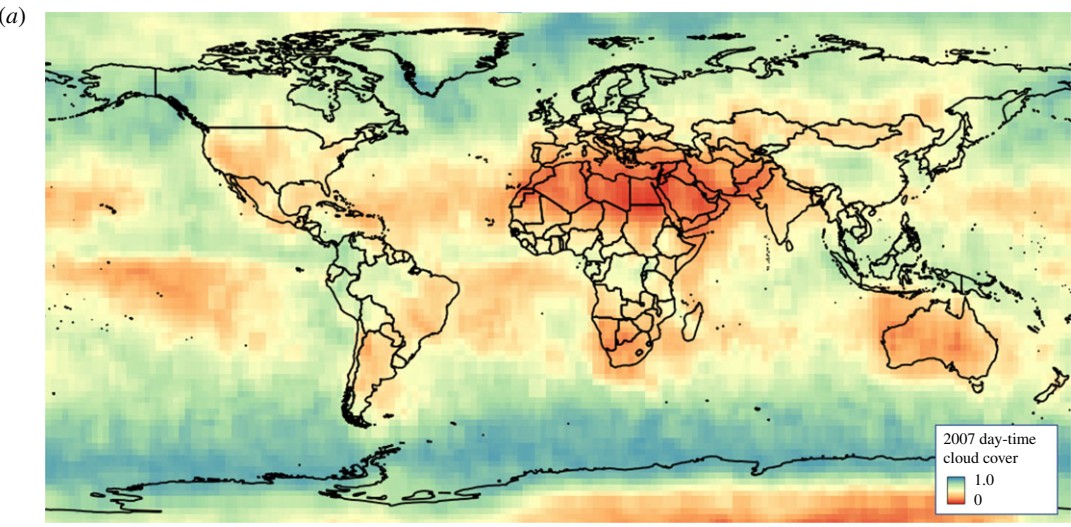

(b)

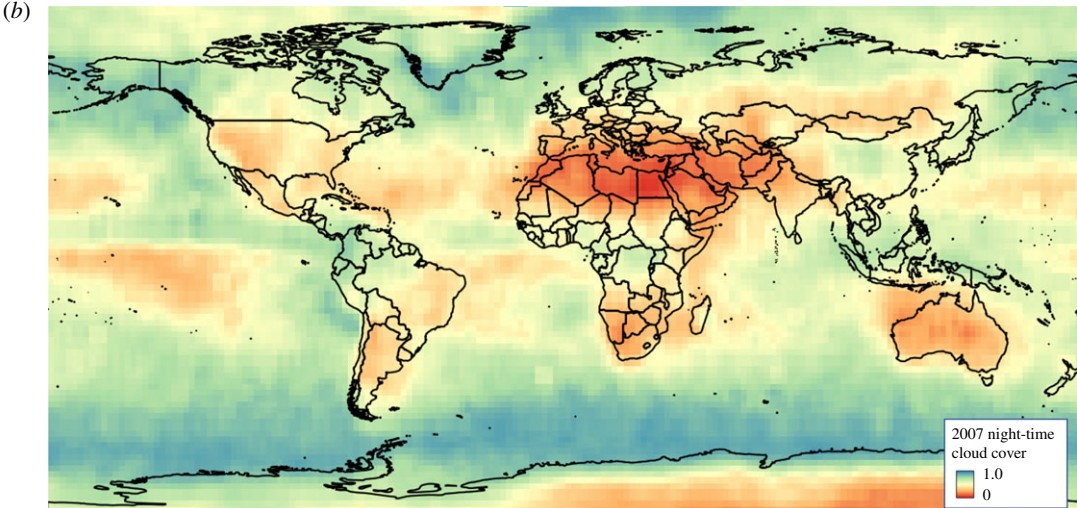

**Figure 4.** Mean cloud cover for 2007 derived from CALIPSO at 2° resolution for day (a) and night (b).

overpasses for a given probability of cloud cover, $c_{frac}$. Using the binomial distribution, this can be solved in terms of the number of overpasses needed, $N_o$,

$$N_o = \frac{\ln(1 - p_{obs})}{\ln(c_{frac})}. \tag{2.7}$$

The probability of cloud cover, $c_{frac}$, was evaluated using the 2007 data from Cloud-Aerosol Lidar and Infrared Pathfinder Satellite Observation (CALIPSO) Cloud-Aerosol LIdar with Orthogonal Polarization (CALIOP) instrument [23], also shown in table 1. The CAL_LID_L3_GEWEX_Cloud-Standard-V1-00 [51] product was acquired for day and night for each month, and the probability of penetration of cloud cover calculated on a 1° grid between 80° S and 80° N averaged over the year for day and night. This has a lot of gaps, particularly in the equatorial region, so it was resampled onto a 2° grid, and the QGIS 'Fill NoData' routine applied. A 3 × 3 convolution filter was applied to remove high-frequency noise, and the results were produced in GeoTIFF format (figure 4). The area-averaged global cloud cover over land was 54.4% for daytime and 56.4% at night, compared with 66% derived from MODIS Aqua (an instrument with the same overpass time as CALIPSO). The difference is most likely due to the different penetration capabilities through thin clouds of passive and active optical systems [52]. To test the interannual variability of cloud cover and provide evidence that 2007 was representative, MODIS Aqua 2.5° mean cloud cover between 2007 and 2016 was calculated. This showed a mean of 66.3% in 2007, and 66.4% over 2007–2016 with a standard deviation of 0.2%, and so the mean cloud cover of 2007 is representative of this 9 year period.

The number of overpasses needed can be calculated in one of three ways: for the global average, per 2° pixel in figure 4 or for a particular fraction of the Earth's surface by finding the cloud cover fraction for a given percentile from a histogram. As it is impossible to have a 100% chance of a cloud-free observation from equation (2.7), no matter how many satellites are used, for this paper a constellation that gives an 80% chance of at least one cloud-free observation for each point on the land at mean cloud cover was considered, echoing the design of GEDI [2]. This requires the mean global cloud cover over land, which was found to be 55% from figure 4. From equation (2.7), at least 2.7 overpasses over each point on the ground would then be needed. Note that this gives an 80% probability of at least one cloud-free observation for any point at the mean cloud cover, but this probability will not be uniformly distributed around the world. Areas of higher cloud cover will have a lower probability of a cloud-free observation, while areas of low cloud cover will have higher probabilities. Analysis in future papers will make use of the spatial detail in figure 4 and account for orbital overlaps to compare potential constellation configurations.

## 2.4.2. Orbits

For a given swath width and altitude, the time for a single satellite to overpass every point on the Earth can be calculated for a given orbital inclination. Combined with the cloud cover, this can be used to calculate the total number of satellites needed for global coverage within a given time-frame.

In order to minimize the number of spacecraft needed, a hybrid constellation with both polar and inclined orbits is investigated. For the spacecraft in inclined orbits, the maximum and minimum latitude, $\delta$, visible to the platform will be $\delta = \pm i$ for a prograde orbit and $\delta = \pm 180 - i$ for a retrograde orbit, where $i$ is the satellite inclination in degrees. The minimum revisit time will occur at latitudes where $\delta = \pm i$. Polar satellites can be used to complement the inclined satellites and provide coverage to latitudes outside these bounds.

To size the constellation, the circumference of the Earth at the maximum latitude band viewable by the inclined satellites, $c$, is calculated as Lowe et al. [53]

$$c = 2\pi \cos(i)\sqrt{\frac{R_{\text{Eq}}^2}{1 + \sin(i)[(1/(1-f)^2) - 1]}}, \tag{2.8}$$

where $R_{\text{Eq}}$ is the equatorial Earth radius, taken as 6378 km, and $f$ is the Earth flattening factor taken as $3.3528 \times 10^{-3}$ from the World Geodetic System 1984, Decker et al. [54]. The number of passes required by polar orbiting spacecraft to completely cover this circumference is $c/s$, where $s$ is the swath width. Assuming that the spacecraft are in circular orbits, the time required for a single orbit period, $T$, can be calculated as

$$T = 2\pi \sqrt{\frac{(R_E + h)^3}{\mu}}, \tag{2.9}$$

where $R_E$ is the mean Earth radius, taken as 6371 km, $\mu$ is the standard Gravitational parameter of the Earth, taken as $3.986 \times 10^{14} \, \text{m}^3 \, \text{s}^{-2}$, and $h$ is the spacecraft altitude. Knowing this, the number of passes each spacecraft will complete per year, $P_y$ is calculated as

$$P_y = 2\frac{3.154 \times 10^7}{T}, \tag{2.10}$$

where $3.154 \times 10^7$ is the number of seconds in a year and the multiplication factor of two accounts for the two passes of the latitude band in a single orbit (i.e. an upwards and a downwards pass).

The number of polar orbiting spacecraft required, $N_p$, can hence be calculated as

$$N_p = \left\lceil \frac{c_{\delta=i}}{sP_y t_{\text{yr}}} \frac{\ln(1 - p_{\text{obs}})}{\ln(c_{\text{frac}})} \right\rceil, \tag{2.11}$$

where $c_{\delta=i}$ is the circumference of the circle drawn by the line of latitude at $\delta = i$, calculated using equation (2.8), $s$ is the instrument swath, calculated using equation (2.5), $c_{\text{frac}}$ is the cloud fraction, and $t_{\text{yr}}$ is the desired time to full coverage in years. If $N_p$ is a fraction, it is rounded up to the nearest whole number to obtain the number of spacecraft.

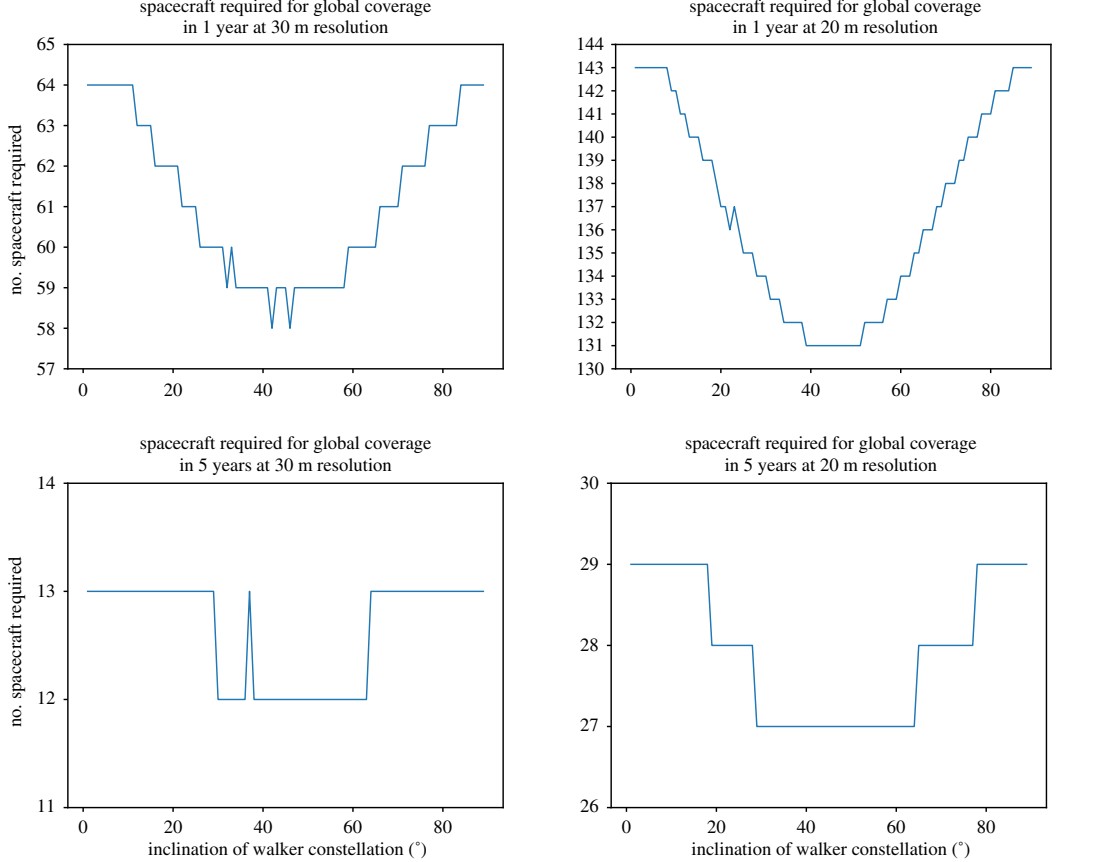

**Figure 5.** Number of satellites required to achieve global coverage at different spatial and temporal resolutions for a mix of polar and inclined satellite orbits.

Given the number of polar orbiting spacecraft required, the coverage provided by these spacecraft at the equator can be calculated as

$$c_p = N_p s P_y t_{yr} \frac{\ln(c_{frac})}{\ln(1 - p_{obs})}. \tag{2.12}$$

The number of spacecraft in inclined orbits required to cover the remaining portion of the equator can then be calculated. Owing to the orbit tilt relative to the equator, the width of the equator seen by the inclined spacecraft on each pass, $W$, is larger than the swath and is calculated as $W = s/\sin(i)$. Hence, the number of inclined spacecraft required, $N_i$, is calculated as

$$N_i = \left\lceil \frac{c_{eq} - c_p}{W P_y t_{yr}} \frac{\ln(1 - p_{obs})}{\ln(c_{frac})} \right\rceil. \tag{2.13}$$

As with $N_p$, $N_i$ is rounded up to the nearest whole number. The number of total spacecraft required to provide global coverage in a given time is then $N_{total} = N_i + N_p$. Note that this assumes no overlap between passes.

Assuming a swath of 300 m per spacecraft, and a mean cloud cover fraction of 55%, the number of total spacecraft required is calculated considering a range of inclinations for the inclined spacecraft. This is seen in figure 5 for a variety of resolutions and times to full coverage. The number of spacecraft required was found to, generally, decrease as the inclination of the inclined satellites approaches of 45°; limited exceptions to this occur due to the fact that whole numbers of spacecraft are required in both the polar and inclined orbits, resulting in occasional spikes as these changes occur. Further reductions could be possible through the inclusion of additional orbits at intermediate inclinations; however, this would further increase the number of launches required, and hence the cost, for the system. It should be noted that as the lidar would only be expected to operate over land, it may be possible to reduce the number of spacecraft required by considering those needed for

**Table 3.** Number of 12 W output lasers (300 m swath) required to achieve global coverage within a given time at a given resolution.

| time to cover | 5 m | 10 m | 20 m | 30 m |
| --- | --- | --- | --- | --- |
| 1 years | 2086 | 522 | 131 | 58 |
| 2 years | 1043 | 261 | 66 | 29 |
| 3 years | 696 | 174 | 44 | 20 |
| 4 years | 522 | 131 | 33 | 15 |
| 5 years | 418 | 105 | 27 | 12 |

full land coverage only. This is proposed as a future development of this work, using a numerical model, with the results proposed here as an upper bound on constellation size.

Using the above method, the minimum possible number of satellites (i.e. using a combination of polar and 45° inclined orbits) to provide coverage for a range of times and resolutions is calculated and given in table 3. It can be seen that annual coverage at 5 m resolution would require the very large number of 2086 spacecraft in orbit. However, global coverage at 30 m resolution once every 5 years, still a significant improvement over what is currently available, could be achieved by 12 spacecraft in orbit, increasing to 27 for a resolution of 20 m.

The number of spacecraft required for global coverage at a given resolution in a given time will vary with output laser power per satellite (the product of payload power and laser efficiency). This is shown in figure 6 for coverage in 1, 2, 3 and 5 years, and an output laser power up to 80 W per spacecraft. The number of spacecraft given are the minimum number possible as calculated using the method described above. Increasing the spacecraft power will reduce the number of spacecraft required as the swath for each will be greater, but this occurs with diminishing returns.

The analysis so far has ignored the issues of duty cycle. If the satellite platform is not capable of powering the lidar at all times, due to loss of solar power at night or heat dissipation limits, it will need to be powered off. In that case, the number of satellites would need to be increased by the same fraction as the loss of duty cycle (e.g. a duty cycle of 50% would require the numbers in table 3 to be doubled). This level of analysis is beyond the scope of this paper but should be accounted for in future work. This being said, if the satellite platform cannot power the lidar at 100% duty cycle, it may be possible to schedule laser downtime to align with periods when overhead sea, clouds or areas that have already been measured [55].

## 2.5. Rough cost for a global lidar system

From table 3, it can be seen that 12 satellites with 12 W output laser power would be needed to produce a 30 m resolution continuous lidar product once every 5 years. This would be equivalent to 12 ICESat-2 satellites in orbit. Multiplying the ICESat-2 mission costs by 12 gives a rough order of magnitude indication of the total cost of a GLS of around $9.1 billion (assuming $760 million for ICESat-2), a fraction of the cost of a global ALS survey (roughly £42 billion). Note that this cost estimate is pessimistic as ICESat-2's cost includes science products and, as it is a one-off, does not have the economy of scale that a GLS could have. A map at 5 m resolution within 5 years would cost roughly $318 billion, exceeding the rough estimate of the cost of a global ALS survey, although the cost per satellite should reduce with the associated mass-production benefits. These costs are perhaps prohibitively expensive, but recent developments in satellites and photonics have the potential to realize a GLS at a more affordable cost.

# 3. Developing an affordable global lidar system

Blair *et al.* [56] point out that improvements in laser efficiency, electrical scanning and deployable optics could enable greater coverage while improvements in small satellites, photonics, signal processing and autonomous on-board systems offer the potential to realize a GLS at a lower cost. Of the improvements suggested by Blair *et al.* [56], only two-position scanning has been implemented so far [2], although that does not solve the power limit of the swath, but allows a rapidly pulsing laser to be scanned across a swath.

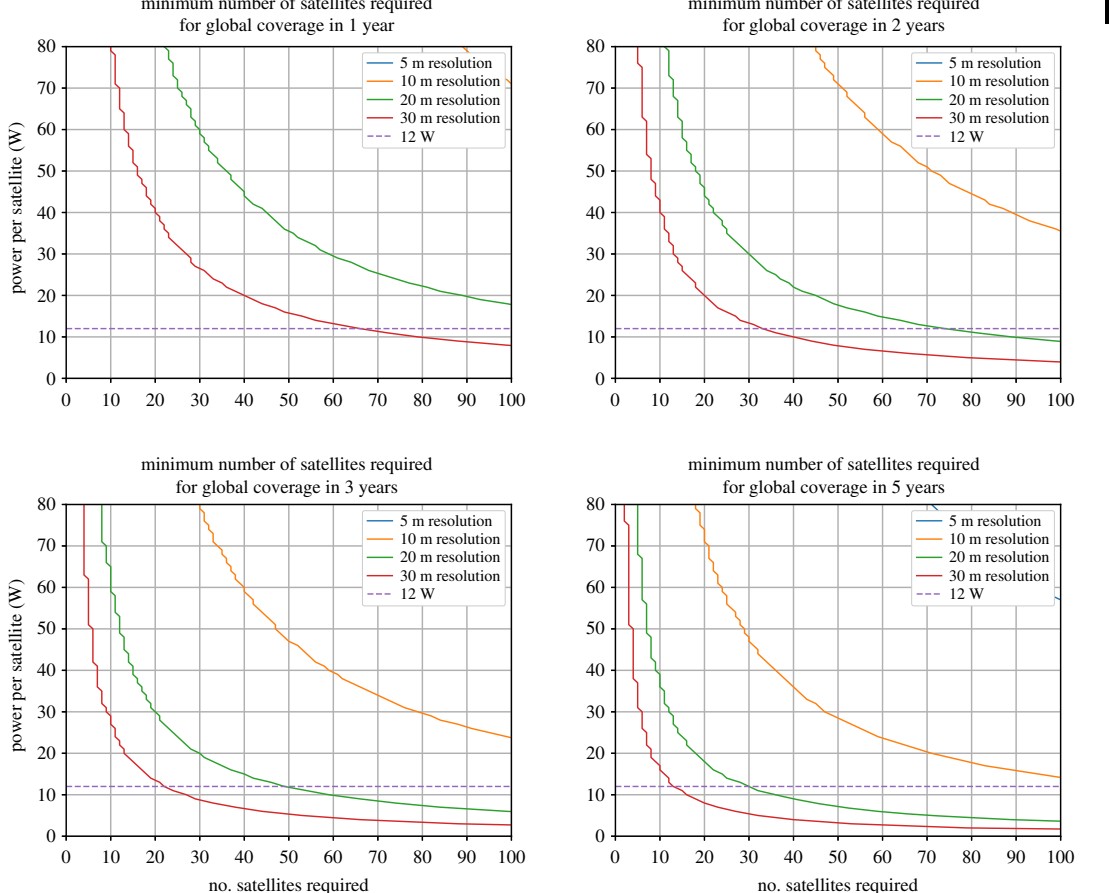

**Figure 6.** Number of satellites in 405 km orbits required to achieve global coverage at different resolutions within a given time-frame against output laser power per satellite. The horizontal dashed line shows the current state-of-the-art 12 W output laser power. GEDI parameters are used for all other characteristics.

Low-cost platforms [57] are revolutionizing access to space. The latest generation of small-sat platforms, such as ClydeSpace's Epic 6U, can provide peak payload powers in excess of 100 W [58] and so it is not impossible to conceive of mounting lidar instruments on low-cost small-sats. Some physical limits would need to be overcome, as a 12U cubesat would have only enough space for a 20 cm diameter telescope; a 16th of the collecting area used in the above analysis. Recent advances in deployable optics should soon allow sufficient collecting area to be unfolded from a small-sat [59]. A lidar small-sat should cost a fraction of the $760 million per platform referenced in §2.5, although a thorough study of the relative costs of different platform size classes should be carried out.

All instruments in table 1 use a Nd:YaG or Nd:YVO$_4$ solid state laser. A Nd:YaG laser has a wall-plug efficiency of 5–8% [36], though LIST proposes increasing this to 15% [34] and an 11% efficient Yb:YAG laser has been reported [37]. Alternative laser sources may be able to provide a similar output power at higher efficiencies, although this has not yet been attempted from a satellite and will require development to reach a sufficient technology readiness level. If achieved, this would increase $L_e$ in equation (2.5), and so the swath width, without requiring more satellite power.

The analysis above assumed a 45% detector efficiency, as used on GEDI [48,60], while ICESat-2 has an efficiency of only 15% [61]. Recent advances in low dead-time, low-noise photon-counting detectors may offer efficiencies of up to 70% [37,62], although they have not yet been flown in space. Fields *et al.* [62] are working towards testing a 70% efficient photon-counting detector onboard a 3U cubesat with ground-based lasers. This could increase $Q$ in equation (2.5), and thus the swath width.

This analysis was based on the minimum detected energy per pulse calculated for GEDI of 0.281 fJ (1505 photons) needed to penetrate 98% canopy cover by night and 96% by day. That has been calculated for an algorithm that processes each footprint in isolation. A GLS would have continuous coverage and so information from adjacent footprints could be used to inform the ground finding algorithm. It has been shown that using such information can constrain the algorithms and allow the ground to be successfully

detected in individual footprints where the beam sensitivities are less than the local canopy cover [49,50]. Applying these techniques may allow $E_{det}$ in equation (2.5) to be reduced, increasing the swath width.

Section 2.4 calculated the number of overpasses needed to give an 80% probability of a cloud-free observation at mean cloud cover. For a GLS with an aim to characterize Earth's clouds, clouded observations would be useful data, but for a GLS aimed at mapping only the land surface, that loss to clouds increases the number of satellites needed (here by a factor of 2.7). Recent advances in autonomous onboard satellite control may allow clouds to be detected and either steered around (if small enough), or else the duty cycle scheduled [55]. The former would effectively reduce $c_{frac}$ in equation (2.7) and so reduce the number of satellites needed, while the latter may allow a reduction in duty cycle below 100% to be mitigated.

All of the above possible developments would allow each satellite to either have a greater swath width, reducing the total number needed, or a lower cost per platform, and so allow a GLS to be realized for a lower cost than estimated in §2.5.

# 4. Conclusion

Past studies have shown that lidar is the optimum technology for measuring bare-Earth elevation in vegetated or topographically complex areas. For this reason, a number of government agencies and companies pay to have ALS data collected over their areas of interest. These ALS surveys have a high cost per unit area and would be prohibitively expensive to perform globally (roughly £41 billion from scaling the cost of Wales' recent countrywide survey). Spaceborne lidars, a number of which have been launched in the last few years and are successfully operating, offer lower cost per unit area, but energy limitations result in sparse coverage; too sparse to be used by many applications that currently rely on ALS data. For spaceborne lidar to be applied to many common ALS data uses, such as flood modelling, it would need to be able to provide continuous cover and produce wall-to-wall maps in order to produce a global lidar system (GLS).

This paper has derived the equations for calculating the number of satellites with a given set of properties needed to achieve global coverage at a given spatial and temporal resolution, accounting for cloud. These were used to calculate how existing in-orbit lidar technology could be scaled up to produce a continuous global map. Propagating the characteristics of existing spaceborne lidars suggests that a lidar satellite aiming to produce a 30 m resolution map with a mean beam sensitivity of 98% by night (and so able to see through dense forest canopies) could have a swath width of 300 m, achieving global coverage (80% of at least one cloud-free observation per ground pixel for mean cloud cover) within 5 years with a constellation of 12 satellites. The equations presented have been put into a python script that is accessible from [63]. Note that the 80% probability of a cloud-free observation is not uniformly distributed around the world and so the spatially explicit cloud map in figure 4 should be used to test coverage at any particular area of interest.

Simply multiplying the cost per satellite for currently in-orbit lidars to put 12 in space would be prohibitively expensive compared with existing civilian remote sensing missions, although a fraction of the cost of a global ALS survey. In addition to economies of scale, recent developments in photonics, lidar signal processing, small satellites and deployable optics should allow a GLS to be realized at a fraction of the cost of existing spaceborne lidars, once they have reached a sufficient technology readiness level.

Data accessibility. Data and relevant code for this research work are stored in GitHub: https://github.com/sthancock/gls_planner and have been archived within the Zenodo repository: https://doi.org/10.5281/zenodo.5579101.

Authors' contributions. S.H. conceived the study and performed the lidar analysis. I.D. provided the cloud and surface reflectance data. C.M. and C.L. performed the platform and orbital analysis. I.W. contributed to the writing, particularly any mention of radar. All authors contributed to the writing.

Competing interests. We declare we have no competing interests.

Funding. This work was funded by the UK Space Agency's National Space Innovation Programme, grant no. NSIP20_N08.

Acknowledgements. Thank you to James Morris, Gerald Bonner and Ludwig Prade for the useful conversations that led to this paper, and thank you to Kristina Tamane for helping to set up the project.

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
