## [Peer Review File · Royal Society Open Science]

Review History

RSOS-211166.R0 (Original submission)

Review form: Reviewer 1

Is the manuscript scientifically sound in its present form?

Yes

Are the interpretations and conclusions justified by the results?

No

Is the language acceptable?

Yes

Do you have any ethical concerns with this paper?

No

Have you any concerns about statistical analyses in this paper?

No

Recommendation?

Reject

Comments to the Author(s)

This is a very interesting paper that discusses optimal satellite number requirement for measuring global bare-Earth elevation using lidar. Based on a great many assumptions - some seem over simplified - the authors came to the conclusion that “a single lidar satellite could have a continuous swath width of 300 m when producing a 30 m resolution map under current technology”, and that “12 satellites would be needed to produce a continuous map every five years”. This is a very bold statement. While I applaud such efforts from the authors, however, I do not find evidence provided in the paper is sufficient to support their conclusion.

First, the authors align lidar satellite development to real-world applications in hydrology, forestry, and urban mapping. This application-oriented setting is a very good starting point, but I am not sure how the authors conclude on the use of 5-m as an optimal resolution for lidar satellite: indeed, as far as I know, most of them would prefer 1-m resolution or even higher. Thus, I do not think the link between these applications and lidar satellite development is appropriate, or at least not as strong as the authors suggested.

Second, I have huge doubts on the impact from cloud and atmosphere. First, the authors only consider a binary case between cloud and non-cloud and rely on one single atmospheric transmittance value of 80%. However, the real world is much more complicated than this: for example, laser would still be able to penetrate through thin cloud, making more shots available for ground measurement; on the other hand, low-lying aerosol and dust has an adverse effect on final ground detection, reducing the total amount of lidar shots. Second, geographic distribution of clouds was not factored in their subsequent calculation. Fig 4 shows the overall patten with which I agree. However, I would argue the requirement of orbit design and repetition would be completely different between equatorial forests and mid-latitude rangelands. This is very important when designing a global lidar constellation.

Third, there are a great many practical and engineering factors being ignored in the manuscript. This is a bit beyond my expertise, but I am very aware of some issues that have not been fully addressed. While the authors mentioned some and provided some preliminary analysis on them, it is too hasty to draw a highly influential conclusion based on current results.

Review form: Reviewer 2

Is the manuscript scientifically sound in its present form?

Yes

Are the interpretations and conclusions justified by the results?

Yes

Is the language acceptable?

Yes

Do you have any ethical concerns with this paper?

No

Have you any concerns about statistical analyses in this paper?

No

Recommendation?

Accept as is

Comments to the Author(s)

The manuscript is an extended derivation of the key calculations required to estimate the requirements for a global lidar mapping system. The work is organized logically and reads well. Overall, this paper is an excellent contribution clarifying the needs for a next generation lidar constellation. I have no issues with the work – it is technically sound and presented clearly. I note only a few corrections.

P3 L24: change “in” to “from”

P10 L38: repeated “the”

P13 L54: delete “the”

Again, I commend the authors on an excellent contribution and expect it will be well received by the scientific community.

Decision letter (RSOS-211166.R0)

Dear Dr Hancock

The Editors assigned to your paper RSOS-211166 "Requirements for a Global Lidar System: Spaceborne lidar with wall-to-wall" have now received comments from reviewers and would like you to revise the paper in accordance with the reviewer comments and any comments from the Editors. Please note this decision does not guarantee eventual acceptance.

Please submit your revised manuscript and required files (see below) no later than 21 days from today's (ie 08-Sep-2021) date. Note: the ScholarOne system will 'lock' if submission of the revision is attempted 21 or more days after the deadline. If you do not think you will be able to meet this deadline please contact the editorial office immediately.

Please note article processing charges apply to papers accepted for publication in Royal Society Open Science (<https://royalsocietypublishing.org/rsos/charges>). Charges will also apply to papers transferred to the journal from other Royal Society Publishing journals, as well as papers submitted as part of our collaboration with the Royal Society of Chemistry

(<https://royalsocietypublishing.org/rsos/chemistry>). Fee waivers are available but must be requested when you submit your revision (<https://royalsocietypublishing.org/rsos/waivers>).

on behalf of Dr Pablo Gonzalez (Associate Editor) and Miles Padgett (Subject Editor)
openscience@royalsociety.org

Associate Editor Comments to Author (Dr Pablo Gonzalez):

Comments to the Author:

Dear Authors,

Thanks for submitting your article to RSOS. I found that the manuscript to be interesting and potentially able to focus the attention of the remote sensing and forestry community around the many challenges around an specific satellite LiDAR mission.

Also, apologies for the delay in finding reviewers. Now, I have received two expert revision reports about the manuscript. As you will be able to read, the opinions are widely different. One reviewer suggesting publication "as is" and a second one "rejection". Both reviewers and myself consider the paper well-written, logical and well-organized.

I critically assessed the review reports. The main concern seems to be related to limitations in the analysis (particularly incompleteness), which might limit applicability and usefulness for scientific discussion and technical design of future satellite LiDAR missions.

However, in my opinion, the manuscript deserves a major revision, mainly in the form of adding information. The authors should pay careful attention to the comments and criticisms from reviewer #1 by clearly stating the many limitations pointed out.

In particular, two limitations stood out:

- 1) which type of applications or scientific information will be not accesible with a 5 m spatial resolution.
- 2) what will be the impact in the success metric/merit of the design under more complex atmospheric transmissivity scenarios (partially clouded and dust atmopheric columns).

I am confident that by addressing those comments the manuscript will be stronger and more useful to the specialized community. Looking forward to the revised manuscript.

Best regards,
Pablo J. Gonzalez

Reviewer comments to Author:

Reviewer: 1

Comments to the Author(s)

This is a very interesting paper that discusses optimal satellite number requirement for measuring global bare-Earth elevation using lidar. Based on a great many assumptions - some

seem over simplified - the authors came to the conclusion that “a single lidar satellite could have a continuous swath width of 300 m when producing a 30 m resolution map under current technology”, and that “12 satellites would be needed to produce a continuous map every five years”. This is a very bold statement. While I applaud such efforts from the authors, however, I do not find evidence provided in the paper is sufficient to support their conclusion.

First, the authors align lidar satellite development to real-world applications in hydrology, forestry, and urban mapping. This application-oriented setting is a very good starting point, but I am not sure how the authors conclude on the use of 5-m as an optimal resolution for lidar satellite: indeed, as far as I know, most of them would prefer 1-m resolution or even higher. Thus, I do not think the link between these applications and lidar satellite development is appropriate, or at least not as strong as the authors suggested.

Second, I have huge doubts on the impact from cloud and atmosphere. First, the authors only consider a binary case between cloud and non-cloud and rely on one single atmospheric transmittance value of 80%. However, the real world is much more complicated than this: for example, laser would still be able to penetrate through thin cloud, making more shots available for ground measurement; on the other hand, low-lying aerosol and dust has an adverse effect on final ground detection, reducing the total amount of lidar shots. Second, geographic distribution of clouds was not factored in their subsequent calculation. Fig 4 shows the overall pattern with which I agree. However, I would argue the requirement of orbit design and repetition would be completely different between equatorial forests and mid-latitude rangelands. This is very important when designing a global lidar constellation.

Third, there are a great many practical and engineering factors being ignored in the manuscript. This is a bit beyond my expertise, but I am very aware of some issues that have not been fully addressed. While the authors mentioned some and provided some preliminary analysis on them, it is too hasty to draw a highly influential conclusion based on current results.

Reviewer: 2

Comments to the Author(s)

The manuscript is an extended derivation of the key calculations required to estimate the requirements for a global lidar mapping system. The work is organized logically and reads well. Overall, this paper is an excellent contribution clarifying the needs for a next generation lidar constellation. I have no issues with the work - it is technically sound and presented clearly. I note only a few corrections.

P3 L24: change “in” to “from”

P10 L38: repeated “the”

P13 L54: delete “the”

Again, I commend the authors on an excellent contribution and expect it will be well received by the scientific community.

===PREPARING YOUR MANUSCRIPT===

one version identifying all the changes that have been made (for instance, in coloured highlight, in bold text, or tracked changes);
 a 'clean' version of the new manuscript that incorporates the changes made, but does not highlight them. This version will be used for typesetting if your manuscript is accepted.

===PREPARING YOUR REVISION IN SCHOLARONE===

- Any electronic supplementary material (ESM).
- If you are requesting a discretionary waiver for the article processing charge, the waiver form must be included at this step.
- If you are providing image files for potential cover images, please upload these at this step, and inform the editorial office you have done so. You must hold the copyright to any image provided.
- A copy of your point-by-point response to referees and Editors. This will expedite the preparation of your proof.

- Ensure that your data access statement meets the requirements at <https://royalsociety.org/journals/authors/author-guidelines/#data>. You should ensure that you cite the dataset in your reference list. If you have deposited data etc in the Dryad repository, please include both the 'For publication' link and 'For review' link at this stage.
- If you are requesting an article processing charge waiver, you must select the relevant waiver option (if requesting a discretionary waiver, the form should have been uploaded at Step 3 'File upload' above).
- If you have uploaded ESM files, please ensure you follow the guidance at <https://royalsociety.org/journals/authors/author-guidelines/#supplementary-material> to include a suitable title and informative caption. An example of appropriate titling and captioning may be found at https://figshare.com/articles/Table_S2_from_Is_there_a_trade-off_between_peak_performance_and_performance_breadth_across_temperatures_for_aerobic_scope_in_teleost_fishes_/3843624.

Author's Response to Decision Letter for (RSOS-211166.R0)

See Appendix A.

Decision letter (RSOS-211166.R1)

Dear Dr Hancock,

It is a pleasure to accept your manuscript entitled "Requirements for a Global Lidar System: Spaceborne lidar with wall-to-wall" in its current form for publication in Royal Society Open Science. The comments of the reviewer(s) who reviewed your manuscript are included at the foot of this letter.

If you have not already done so, please ensure that you send to the editorial office an editable version of your accepted manuscript, and individual files for each figure and table included in

your manuscript. You can send these in a zip folder if more convenient. Failure to provide these files may delay the processing of your proof.

on behalf of Dr Pablo Gonzalez (Associate Editor) and Miles Padgett (Subject Editor)
openscience@royalsociety.org

Associate Editor Comments to Author (Dr Pablo Gonzalez):

Associate Editor

Comments to the Author:

Dear authors,

The response of the authors to the reviewers and associated editor has addressed to reasonable level the comments of reviewer number 1, and particularly the two comments I had highlighted.

I consider that this manuscript, after the revision addressing the comments of reviewer number 1, has improved the content and merit of the manuscript. Therefore, I concluded that the manuscript has to be published as it is.

Best regards,
Pablo J. Gonzalez

Reviewer comments to Author:

Appendix A

Thank you to both reviewers and the editor for their comments.

Editor

1) which type of applications or scientific information will be not accesible with a 5 m spatial resolution.

To address this, section 2 has been modified to:

"The number of footprints, and so laser pulse rates and power, is controlled by the required resolution, which in turn is controlled by the laser footprint size. The resolution requirement depends upon the application but there are some limits. It has been shown that too large a footprint will prevent the ground being seen underneath a forest due to overlap between the ground and vegetation returns~\cite{Harding_2005,Hancock_2012}. For example, ICESat cannot be used to measure forest height on slopes steeper than $10\text{--}12^\circ$ ~\cite{Los_2012} because of its large (65–90 m diameter) footprint. An upper footprint diameter limit of around 30 m has been suggested~\cite{Dubayah_2020}. There is no lower limit to the resolution, until the diffraction limit (which, for example, is 65 cm for GEDI's 0.5 m~\(^{2}\)) telescope at an altitude of 405 km using a 1064 nm wavelength), but as the laser footprint gets smaller, the number of pulses needed to achieve the sampling density required to achieve the higher resolution increases and so the total power requirement is greater. The National Research Council's decadal survey has set a minimum resolution of 5 m for a bare-Earth DEM~\cite{NRC_2007}, but it should be noted that a global bare-Earth DEM of even 30 m resolution would revolutionise a number of fields. Some applications require very high spatial-resolution, such as detecting small dwelling foundations for archaeology and mapping street furniture~\cite{Banaszek_2018}. These tend to require multiple (10+) pulses per square metre, which would require resolutions of a few tens of centimetres and so be very challenging to achieve with a satellite lidar given the technology likely to be available over the next few years. In addition, at those spot sizes the system would suffer significant geolocation issues from laser jitter~\cite{Luthcke_2019}. As many applications would benefit from 5–30 m resolution data, especially vegetation studies~\cite{Dubayah_2020}, flood modelling~\cite{Courty_2019} and some geomorphological processes~\cite{Grieve_2016}, this paper will focus on systems with the less energy intensive spatial resolutions of 5–30 m.

For the minimum temporal resolution, there has never been a global bare-Earth DEM, so any time-scale is an improvement over what is currently available. For those countries that have continuous ALS coverage, the data has never been collected more regularly than once every ten years, so this could provide an upper limit to the time to coverage.

This paper will investigate the requirements for producing a global map with between 5 and 30 m resolution, completed once every one to five years. Five years has been chosen rather than ten as satellite lidars have so far only been designed for 3-6 years of operation, although CALIPSO has far exceeded that. It should be noted that the equations presented can be used for any spatial-resolution product and the preliminary analysis did consider 1 m resolution, but the number of spacecraft required were already so large at 5 m resolution that the results were only presented for 5-30 m resolution."

More specific replies are given below in the response to reviewer 1. We hope that this addresses this point?

2) what will be the impact in the success metric/merit of the design under more complex atmospheric transmissivity scenarios (partially clouded and dust atmospheric columns).

This paper focuses on a satellite constellation capable of maintaining a mean beam sensitivity of 98%. This was the approach used in the planning of the GEDI mission and so it has flight heritage. The reviewer is correct that the atmospheric transmissivity will vary in space and time and will be lower than 80% for 50% of the time. That data will have a beam sensitivity lower than 98%, but it will only lead to a measurement error if the local canopy cover is higher than the beam sensitivity. For this reason, an accurate map of canopy cover is needed to determine whether a lower atmospheric transmission will lead to a measurements error. Passive optical maps of canopy cover saturate at 70%, well below the values of interest here (90-98%). The only reliable way to map canopy covers of 90-98% is with GEDI, but the L2B product is not yet dense or mature enough (having not yet been validated by the product team in a peer reviewed paper) to allow this analysis to take place. For these reasons, we hope that echoing the instrument requirements of a successful mission with flight heritage will be acceptable for this analysis of an energy limited system?

To make this clear, section 2 c has been modified to:

"From table~\ref{TABsystems}, the current highest output power free-flying lidar satellite is ICESat-2 at 12 W (CATS is mounted on the ISS). Note that higher output powers than this may be possible~\cite{Yu_2010,Abshire_2020}, but for this paper the highest demonstrated lidar output power will be used as an upper limit. Using the $(P_{\text{pay}}.L_{\text{e}}=12 \text{ W})$ from ICESat-2 and $(E_{\text{det}}=0.281 \text{ fJ})$, $(A=0.5 \text{ m}^2)$, $(Q=45\%)$, $(h=405 \text{ km})$, $(\tau=0.8)$ from GEDI, the swath widths for different resolutions are shown by the curve in figure~\ref{FIGswathSat}. $(\tau=0.8)$ is chosen as an average value to give an average returned energy and so the average beam sensitivity of 98% by night. It should be noted that atmospheric transmission varies in space and time, but this paper explores what constellation would be

required to maintain a mean beam sensitivity of 98% by night, echoing the method used for the planning for GEDI~\cite{Dubayah_2020,Hancock_2019}. The beam sensitivity will vary as the atmospheric transmission varies and some additional data may require filtering out if the beam sensitivity falls below the local canopy cover, although a continuous raster of footprints should allow that to be offset by more advanced signal processing~\cite{Zhou_2017,Neuenschwander_2019}."

Thank you to both reviewers for their comments.

Reviewer 1

This is a very interesting paper that discusses optimal satellite number requirement for measuring global bare-Earth elevation using lidar. Based on a great many assumptions - some seem over simplified - the authors came to the conclusion that "a single lidar satellite could have a continuous swath width of 300 m when producing a 30 m resolution map under current technology", and that "12 satellites would be needed to produce a continuous map every five years". This is a very bold statement. While I applaud such efforts from the authors, however, I do not find evidence provided in the paper is sufficient to support their conclusion.

Thank you for the comment. The fundamental limitation to lidar coverage is the amount of energy it can emit and still make an accurate measurement per footprint (cloud cover permitting, which is covered below). That is why the coverage of a lidar is so much sparser than a passive optical or a radar system. How those energy requirements affect the achievable swath width can be exactly calculated by equation 2.5. No assumptions are needed to reach a conclusion of how many satellites are needed to give a mean beam sensitivity of 98% by night (96% by day) with a probability of at least one-cloud free observation per point on the ground of 80% at mean cloud cover. It is a combination of the well-established lidar equation and the orbital ground-speed equation. We then put in the known characteristics of in-orbit lidars (telescope size, output laser power, detected energy etc.) and from that arrived at the 300 m swaths. It is not clear to me what other assumptions the reviewer thinks makes the calculation of a 300 m swath unrealistic?

An alternative, but less generalisable, way to calculate it is to scale up the existing technology, and we have done this as a sanity check. GEDI has three lasers which emit 4 footprints simultaneously. If these were focused to give 30 m footprints (rather than the current 22 m, and there is no reason they could not be) and aligned to be adjacent to each other (again no reason they could not), that would be a continuous swath of 120

m. That is with a laser output power of 7.3W, with the lasers split into 2*power beams (99% beam sensitivity) and 2*coverage beams (98% beam sensitivity). Increasing the output power to the achievable 12W of ICESat-2 and splitting all beams into coverage beams ($E_{det}=0.281fJ$, giving 6*30 m footprints) gives a swath of:

$$\text{swath} = 6*30*12/7.3 = 295.89 \text{ m}$$

The extra power of 12W over GEDI could either be used to pulse the existing laser faster (if the technology allows), to produce more energetic pulses per laser to be subsequently split (if the technology allows), or to power additional lasers (if the technology does not allow). The energy requirements are identical for all approaches. So we can analytically calculate the 300 m swath from the currently in-orbit technology, without making any assumptions. There is no technological barrier to refocusing GEDI's beams to give 6 coverage beams in a continuous swath and no reason a solid-state laser could not output 12 W from a satellite, especially at the coverage beam pulse energies of ~ 5mJ. I admit that there could be optical element heating threshold damage issues at higher powers for a single laser beam and there are limits to that, but those are not an issue at the powers being considered here (as proven by the data being returned by GEDI and ICESat-2), and, if it were essential, could be designed around by using larger optical elements and wider laser beams to reduce the local intensity, or using multiple lasers with less power per beam to achieve the required swath (rather than one higher power laser split into multiple beams).

So, I would argue that there is no reason to doubt that a 300m swath at 30 m resolution could be achieved and we have shown that from first principles, with an equation that allows the analysis to be generalised to any satellite configuration and end user need. As far as we aware this generalisation has not been presented before (and reviewer 2 appears to agree)?

Next, we used that to calculate the number of satellites needed to give a set probability of a cloud-free observation. Again, I am not sure what assumptions the reviewer is referring to? Satellite orbits are predictable, and so, for a given swath width, we can calculate exactly how often it will pass over a point within a given timeframe. The only factor we need to take into account is the cloud cover. We have measured the average lidar perceived cloud cover over the whole world from CALIPSO (which accounts for lidar penetrating thin haze etc.). Once the average cloud cover is known, the binomial equation can be used to determine how many overpasses are needed to give a certain probability of at least 1 cloud-free observations (equation 2.7). If our CALIPSO cloud cover map is accurate (and it should be noted that this same data source was used for the planning of the GEDI mission and was accepted by multiple NASA review panels), then we think it is uncontroversial that equation 2.7 can be combined with the orbital equations to calculate the number of satellites needed to give an 80% chance of at least one cloud free observation for the average cloud cover (more on this in answer to your later question)?

Putting in the 300m we proved from first principles (and again by scaling GEDI and ICESat-2 above) and the CALIPSO cloud map, we get 12 satellites to have an 80% chance of at least one cloud free observation per point on the ground at the average cloud cover. As stated later, we accept that this is not a uniform probability over the whole world, but it is the probability at mean cloud cover.

The wording has been adjusted in the conclusions and in a few other places to make it clear that our aim is to have a lidar constellation with a mean beam sensitivity of 98% by night (see later answer for full details) and to have an 80% chance of at least one cloud-free observation per point on the ground at a mean cloud cover of 55% (see later answer for full details and acknowledgement that this 80% probability will not be uniform). You are correct that if a specific application has a specific need for data coverage at a specific place, a more detailed analysis would be possible, and this has been made clear in the conclusions.

First, the authors align lidar satellite development to real-world applications in hydrology, forestry, and urban mapping. This application-oriented setting is a very good starting point, but I am not sure how the authors conclude on the use of 5-m as an optimal resolution for lidar satellite: indeed, as far as I know, most of them would prefer 1-m resolution or even higher. Thus, I do not think the link between these applications and lidar satellite development is appropriate, or at least not as strong as the authors suggested.

I agree that 1 m would be much better than 5 m resolution. Some applications of ALS even ask for 10 cm resolution, such as most archaeology (eg. building foundation outlines, but not temples in forests) and street furniture. We considered 5 m as a lower limit for four main reasons.

1) First there is no bare-Earth terrain map or directly measured canopy height map available globally at any resolution. Some countries do have 1m-10 cm resolution ALS data available, but many do not and what ALS data there is is not frequently updated. All of those applications, such as flood modelling and urban mapping, in countries without access to ALS derived bare-Earth DTMs would benefit from a DTM at any resolution.

2) Our preliminary analysis did go down to 1 m resolution, but the numbers of satellites needed became very large. As the estimated cost for 5 m resolution was already many orders of magnitude beyond existing missions, we chose to truncate the results presented to 5 m. The equations and python tool presented can have 1 m resolution input if a user wants to investigate it, but for a pure satellite lidar that would perhaps be prohibitively expensive with technology likely to become available over the next few years. This has now been made clear in the paper.

3) NASA's Surface Topography & Vegetation report, aiming to make a map of bare-Earth elevation and tree height, had chosen 5 m as their aim (<https://science.nasa.gov/earth-science/decadal-stv>). Interestingly they did not define a temporal resolution and so one aim of this analysis was to fill that gap, as well as to provide a simple tool to assess how lidar constellations of different configurations could address that need.

4) My own area of interest is primarily forests. For forests, lidar satellites are only ever likely to provide stand scale properties (unless we can get down to 10 cm resolution) and so footprints need to be large enough to cover at least one tree (see GEDI papers cited) and so for those applications, 5 m is more than adequate and 30 m often used.

Currently not even a 30 m bare-Earth DTM nor a CHM are available globally yet. DSMS are (such as SRTM, which has been used for flood modelling despite being a DSM), but not DTMs and CHMs. Even a 30 m res DTM would be revolutionary.

To address this, section 2 has been modified to:

"The number of footprints, and so laser pulse rates and power, is controlled by the required resolution, which in turn is controlled by the laser footprint size. The resolution requirement depends upon the application but there are some limits. It has been shown that too large a footprint will prevent the ground being seen underneath a forest due to overlap between the ground and vegetation returns~\cite{Harding_2005,Hancock_2012}. For example, ICESat cannot be used to measure forest height on slopes steeper than 10-12\(^{\circ}\)~\cite{Los_2012} because of its large (65-90 m diameter) footprint. An upper footprint diameter limit of around 30 m has been suggested~\cite{Dubayah_2020}. There is no lower limit to the resolution, until the diffraction limit (which, for example, is 65 cm for GEDI's 0.5 m\(^{2}\) telescope at an altitude of 405 km using a 1064 nm wavelength), but as the laser footprint gets smaller, the number of pulses needed to achieve the sampling density required to achieve the higher resolution increases and so the total power requirement is greater. The National Research Council's decadal survey has set a minimum resolution of 5 m for a bare-Earth DEM~\cite{NRC_2007}, but it should be noted that a global bare-Earth DEM of even 30 m resolution would revolutionise a number of fields. Some applications require very high spatial-resolution, such as detecting small dwelling foundations for archaeology and mapping street furniture~\cite{Banaszek_2018}. These tend to require multiple (10+) pulses per square metre, which would require resolutions of a few tens of centimetres and so be very challenging to achieve with a satellite lidar given the technology likely to be available over the next few years. In addition, at those spot sizes the system would suffer significant geolocation issues from laser jitter~\cite{Luthcke_2019}. As many applications would benefit from 5-30 m resolution data, especially vegetation studies~\cite{Dubayah_2020}, flood modelling~\cite{Courty_2019} and some geomorphological

processes~\cite{Grieve_2016}, this paper will focus on systems with the less energy intensive spatial resolutions of 5-30 m.

For the minimum temporal resolution, there has never been a global bare-Earth DEM, so any time-scale is an improvement over what is currently available. For those countries that have continuous ALS coverage, the data has never been collected more regularly than once every ten years, so this could provide an upper limit to the time to coverage.

This paper will investigate the requirements for producing a global map with between 5 and 30 m resolution, completed once every one to five years. Five years has been chosen rather than ten as satellite lidars have so far only been designed for 3-6 years of operation, although CALIPSO has far exceeded that. It should be noted that the equations presented can be used for any spatial-resolution product and the preliminary analysis did consider 1 m resolution, but the number of spacecraft required were already so large at 5 m resolution that the results were only presented for 5-30 m resolution."

Second, I have huge doubts on the impact from cloud and atmosphere. First, the authors only consider a binary case between cloud and non-cloud and rely on one single atmospheric transmittance value of 80%. However, the real world is much more complicated than this: for example, laser would still be able to penetrate through thin cloud, making more shots available for ground measurement; on the other hand, low-lying aerosol and dust has an adverse effect on final ground detection, reducing the total amount of lidar shots. Second, geographic distribution of clouds was not factored in their subsequent calculation. Fig 4 shows the overall pattern with which I agree. However, I would argue the requirement of orbit design and repetition would be completely different between equatorial forests and mid-latitude rangelands. This is very important when designing a global lidar constellation.

It should be noted that the methodology we have used here echoes that used for the NASA GEDI mission, using a single global mean cloud cover and a single mean atmospheric transmittance, and so has flight heritage. We thought that reusing the methodology for a successful mission would allow us to most robustly estimate future mission performance, leading in to a future tradespace study. Admittedly an ISS altitude shift has caused GEDI's coverage to deviate from the pre-launch predictions due to orbital resonance, but until that shift it was matching near perfectly (as presented at the AGU conference), suggesting that the method used here is appropriate.

The cloud map we have derived is an accurate measure of the lidar perceived cloud cover from CALIPSO. Comparing to a MODIS derived cloud cover, the CALIPSO cloud cover is lower (approximately 10%), which accounts for those penetrations through thin clouds you mention. You are

right that there are multiple ways to account for the distribution of cloud cover around the world. As we state in the paper:

"The number of overpasses needed can be calculated in one of three ways; for the global average, per 20 pixel in figure 4, or for a particular fraction of the Earth's surface by finding the cloud cover fraction for a given percentile from a histogram."

In a follow on paper, exploring the possibility of using CubeSats to lower the cost, we have used the spatially explicit method. You are right that some areas are cloudier than others, especially the tropics. Here we have chosen to look at the number of satellites needed to give an 80% probability of seeing the ground through the average cloud cover. This comes out as 2.7 overpasses per point on the ground. You are correct that this configuration would have a lower probability of seeing the ground in areas with much higher cloud cover. For 2.7 orbits the probability of at least one cloud-free observation is 80% if the cloud cover is 55%, but if the mean cloud cover increases to 90% (as in some areas in the tropics), the probability of a cloud free observation for 2.7 overpasses falls to 25%. Here we were looking at the constellation configuration needed to give that 80% probability of cloud free observation at mean cloud cover. The equations are general and can be given any probability of observation or cloud cover values, and future studies looking at applying these to specific applications with specific data needs should take these factors into account.

It should be noted that due to the nature of the binomial distribution, it is impossible to guarantee a 100% chance of a cloud free observation, even at low average cloud covers. Because of this we have used the 80% mean probability of seeing each point on the ground through mean cloud cover (as used on GEDI), and this calculation requires the mean cloud cover. I would argue that we have not made assumptions, but have defined the mission design in relation to the measure of mean cloud cover, as GEDI was.

We took a similar approach with the atmospheric transmissivity, using the global average to determine the energy needed to give a mean beam sensitivity of 98% by night. This was the approach used in the planning for the GEDI mission and so has flight heritage. You are correct that 50% of the signal will be through lower transmission atmospheres, and so have a lower returned energy. However, that only prevents an accurate measurement if it causes the beam sensitivity of the laser shot to be lower than the local canopy cover. A similar spatially explicit or histogram approach discussed for clouds could have been applied here, but it would need a map of canopy cover to estimate amount of data lost. Unfortunately at the moment only GEDI is capable of measuring the high canopy covers, >80%, needed, but the L2B product is not yet spatially dense or mature enough (having not yet been validated by the product team in a peer reviewed paper) to allow this analysis to take place. In future analysis we will explore the spatial distribution of cloud, atmospheric transmission and canopy cover. But I would argue that the results

presented here are accurate in terms of giving a mean beam sensitivity of 98% by night and make no assumptions. The sensitivity will vary, but the mean will be 98%.

The main purpose of this paper was to introduce the method for estimating the total number of satellites needed for global coverage. This method was then illustrated by putting in the parameters for currently in orbit technology. I argue that our statement that a fleet of 12 lidar satellites with the characteristics described would have an 80% probability of viewing each point on the land surface through average cloud cover with an average beam sensitivity of 98% by night is correct. The other factor missed out from this analysis is the overlap of the orbits. There will be overlap towards the edge of each orbital inclination angle, locally increasing the chance of a cloud free observation. For satellites in lower inclination orbits, that can include some of the tropics.

To try and make this more clear, the paragraph at the end of section 2.d.i has been changed to:

"The number of overpasses needed can be calculated in one of three ways; for the global average, per $2^{(o)}$ pixel in figure~\ref{FIGcloud}, or for a particular fraction of the Earth's surface by finding the cloud cover fraction for a given percentile from a histogram. As it is impossible to have a 100% chance of a cloud-free observation from equation~\ref{EQcloud}, no matter how many satellites are used, for this paper a constellation that gives an 80% chance of at least one cloud-free observation for each point on the land at mean cloud cover was considered, echoing the design of GEDI~\cite{Dubayah_2020}. This requires the mean global cloud cover over land, which was found to be 55% from figure~\ref{FIGcloud}. From equation~\ref{EQcloud}, at least 2.7 overpasses over each point on the ground would then be needed. Note that this gives an 80% probability of at least one cloud-free observation for any point at the mean cloud cover, but this probability will not be uniformly distributed around the world. Areas of higher cloud cover will have a lower probability of a cloud-free observation, whilst areas of low cloud cover will have higher probabilities. Analysis in future papers will make use of the spatial detail in~\ref{FIGcloud} and account for orbital overlaps to compare potential constellation configurations."

And this sentence added to section 3:

"Section~\ref{SECorbits} calculated the number of overpasses needed to give an 80% probability of a cloud-free observation at mean cloud cover."

And this to the conclusions:

"Note that 80\% probability of a cloud-free observation is not uniformly distributed around the world and so the spatially explicit cloud map in figure~\ref{FIGcloud} should be used to test coverage at any particular area of interest for particular applications."

We hope that the reviewer can see the benefit of this initial, global average analysis using currently in-orbit technology to give a mean beam sensitivity of 98% and an 80% chance of at least one cloud free observation at mean cloud cover. We shall build upon this in future papers when exploring different satellite configurations. This paper is intended to provide a framework and the benchmark to compare those alternative configurations against. Moving to a spatially explicit analysis is likely to increase the number of satellites needed, and as building 12 ICESat-2 class satellites is already going to be prohibitively expensive, we feel that the steps needed to make a GLS affordable, outlined in section 3, are a higher priority than performing a more spatially explicit analysis which is very likely to return a similar or higher total cost estimate.

Third, there are a great many practical and engineering factors being ignored in the manuscript. This is a bit beyond my expertise, but I am very aware of some issues that have not been fully addressed. While the authors mentioned some and provided some preliminary analysis on them, it is too hasty to draw a highly influential conclusion based on current results.

I am not sure what issues the author is aware of that could prevent these results being true? It is correct that designing satellite lidars is not trivial and there are concerns about output power, element damage thresholds, optical alignment, platform pointing accuracy etc. which must be taken into account during the optical design. However the parameters used in this analysis are taken from in orbit instruments and so are well within the current technology envelope. We have shown from first principles how coverage is related to output laser power (the primary limit for a lidar), and so the optical configuration within the instrument is less important than the factors in equation 2.5. Because of this we stand by these conclusions.

Are you perhaps referring the laser instantaneous powers and how that might damage optical components? If so, that was a concern when building GEDI and caused the output laser energy to be decreased from 14 mJ to 10 mJ, but GEDI has shown that it is possible to build a system that outputs 10 mJ per laser shot (the value used in our analysis), safely transmitted through the optics and maintaining performance for 2.5 years now. We are not considering increasing the instantaneous power, but the pulse repetition rate. Note that from a power usage point of view, it is

irrelevant whether the faster rep rate comes from a single laser pulsing more rapidly or multiple lasers firing in sequence. The power usage will be identical (although the cost and weight of multiple lasers in sequence would be higher). So it would be possible to construct this system by placing multiple GEDI type lasers (which have flight heritage) on the same platform, if it proves that a single laser cannot achieve the pulse repetition rate and pulse energy required to achieve the 12W output power. The limit on coverage is from total power and the equations presented are applicable to any lidar configuration, no matter how you choose to split the output between different laser units/scanning mechanisms etc. We are completing a follow on paper which explores these issues and investigates ways to push the technology envelope in order to maximise coverage per unit cost, but we would argue that the parameters used in this analysis already have flight heritage and so are achievable.

If the reviewer can provide any specific concerns, we would be happy to address them.

Reviewer 2:

#Comments to the Author(s)

#The manuscript is an extended derivation of the key calculations required to estimate the requirements for a global lidar mapping system. The work is organized logically and reads well. Overall, this paper is an excellent contribution clarifying the needs for a next generation lidar constellation. I have no issues with the work - it is technically sound and presented clearly. I note only a few corrections.

#

Thank you for your kind comments.

#

#P3 L24: change "in" to "from"

#

Done

#

#P10 L38: repeated "the"

#

Corrected

#

#P13 L54: delete "the"

#

I am afraid that my page 13 does not have 54 lines, so I am not quite sure which the is being referred to?

#

#Again, I commend the authors on an excellent contribution and expect it will be well received by the scientific community.

#

Thank you very much. That is very nice to hear.

#